# Patient Selection and Outcomes of Laparoscopic Microwave Ablation of Hepatocellular Carcinoma

**DOI:** 10.3390/cancers15071965

**Published:** 2023-03-25

**Authors:** Robert C. G. Martin, Matthew Woeste, Michael E. Egger, Charles R. Scoggins, Kelly M. McMasters, Prejesh Philips

**Affiliations:** The Department of Surgery, Division of Surgical Oncology, University of Louisville School of Medicine, Louisville, KY 40292, USA

**Keywords:** microwave ablation, thermal ablation, hepatocellular carcinoma, liver

## Abstract

**Simple Summary:**

A prospective evaluation of laparoscopic microwave ablation (MWA) of hepatocellular carcinoma is underutilized and predictors of patient selection and survival in this setting are not well characterized. Preoperative single lesions, “pusher” radiographic presentation, and total bilirubin < 2, and AFP < 20 ng/mL independently predict PFS and OS following operative MWA.

**Abstract:**

Background: Laparoscopic microwave ablation (MWA) of hepatocellular carcinoma is underutilized and predictors of survival in this setting are not well characterized. Methods: The prognostic value of clinicopathologic variables was evaluated on progression-free survival (PFS) and overall survival (OS) by univariate and multivariate analyses. The aim of this study was to evaluate a preferred laparoscopic MWA approach in HCC patients that are not candidates for percutaneous ablation and further classify clinicopathologic factors that may predict survival outcomes following operative MWA in the setting of primary HCC. Results: 184 patients with HCC (median age 66, (33–86), 70% male) underwent laparoscopic MWA (N = 162, 88% laparoscopic) compared to 12% undergoing open MWA (N = 22). Median PFS was 29.3 months (0.2–170) and OS was 44.2 months (2.8–170). Ablation success was confirmed in 100% of patients. Ablation recurrence occurred in 3% (6/184), and local/hepatic recurrence occurred in 34%, at a median time of 19 months (9–18). Distant progression was noted in 8%. Median follow up was 34.1 months (6.4–170). Procedure-related complications were recorded in six (9%) patients with one 90-day mortality. Further, >1 lesion, AFP levels ≥ 80 ng/mL, and an “invader” on pre-operative radiology were associated with increased risk of progression (>1 lesion HR 2.92, 95% CI 1.06 –7.99, *p* = 0.04, AFP ≥ 80 ng/mL HR 4.16, 95% CI 1.71–10.15, *p* = 0.002, Invader HR 3.16, 95% CI 1.91–9.15, *p* = 0.002 ) and mortality (>1 lesion HR 3.62, 95% CI 1.21–10.81, *p* = 0.02], AFP ≥ 80 ng/mL HR 2.87, 95% CI 1.12–7.35, *p* = 0.01, Invader HR 3.32, 95% CI 1.21–9.81, *p* = 0.02). Conclusions: Preoperative lesion number, AFP ≥ 80 ng/mL, and an aggressive imaging characteristic (Invader) independently predict PFS and OS following laparoscopic operative MWA.

## 1. Introduction

Hepatocellular carcinoma is the most common primary liver cancer, accounting for 80–90% of diagnoses [1]. Simultaneous chronic liver diseases and a heterogenous stage presentation creates a dual challenge in the management of HCC [2,3]. Therefore, treatment of patients with HCC is often limited by the age of the patient, severity of cirrhosis (Child–Pugh status or Model for End Stage Liver Disease (MELD) or presentation in advanced stages of disease (Barcelona clinic liver cancer (BCLC)). Orthotopic liver transplantation (OLT) and complete tumor resection are the ultimate goals for operative treatment for those with early-stage HCC [4]. However, certain patients cannot tolerate OLT or resection due to concomitant comorbidities despite meeting Milan criteria [5]. Consequently, a significant subset of patients rely on an alternative effective treatment modality—thermal ablation.

Thermal ablation has been increasingly integrated into the treatment algorithm for Barcelona Clinic Liver Cancer (BCLC) stage 0-B HCC patients unable to receive OLT or complete resection. Microwave ablation (MWA) is one modality that generates high-frequency electromagnetic energy deposited directly into the tumor tissue surrounding the antenna probe [6,7,8]. Currently, MWA is most often performed percutaneously [9]; however, not all tumors are accessible by this approach [10]. Laparoscopic MWA of HCC is a safe and efficacious treatment alternative for those with appropriate tumor- or patient-specific indications [11,12]. Despite advances in laparoscopy and improved morbidity, this technique remains underutilized for primary and secondary liver tumors [13].

Few studies have evaluated long-term outcomes of operative or laparoscopic MWA in HCC, proposing low rates of regional recurrence [14,15,16]. Ryu et al. recently identified risk factors of survival in a large cohort of intermediate-stage HCC patients who underwent operative MWA; however, these results stand alone and remain to be validated across other cohorts [17]. Therefore, the intention of this study was twofold. First, to evaluate our experience utilizing a preferred laparoscopic MWA approach in HCC patients that are not candidates for percutaneous ablation. Second, we aimed to further classify clinicopathologic factors that may predict survival outcomes following operative MWA in the setting of primary HCC.

## 2. Materials and Methods

### 2.1. Patient Population

This is a prospective evaluation of a University of Louisville (Louisville, KY, USA), Institutional Review Board (IRB) approved database reviewed for all patients undergoing surgical (laparoscopic or open) MWA for HCC from August 2004 to May 2022. Diagnoses were established by cross-sectional radiographic imaging in correlation with serum alpha fetal protein (AFP) levels or by tissue biopsy [18]. Included into this evaluation were all patients > 18 years of age, with primary HCC amenable to complete ablation, combination of ablation plus bland embolization the day prior, or ablation in conjunction with resection with curative intent. Prior systemic chemotherapy (i.e., tyrosine kinase inhibitors) or locoregional interventions (DEBDOX or Y-90) were allowed and utilized in a neoadjuvant approach to assess the biology of disease (i.e., test of time), evaluate sub-centimeter extra-hepatic disease, or while improving the patient’s overall hepatic function (i.e., abstaining from alcohol, treating hepatitis B or C viral load, or other modifiable co-morbidities). Patients who received percutaneous intervention or MWA for diagnoses other than HCC were not included in this analysis. The selection for the MWA approach, percutaneous or laparoscopic, was agreed upon based on evaluation and consensus of both the interventional radiologist (IR) and surgeon on a case-by-case basis discussed in weekly multi-disciplinary tumor boards and by referral. Initial decision-making bias by our group would be based on (1) proximity to surrounding structures, (2) ease of percutaneous access, (3) ease of laparoscopic access, (4) the need for staging laparoscopic evaluation, and (5) degree of portal hypertension (platelet count and coagulopathy). The intraoperative approach was generally favored in presence of one or more of the following conditions: multi-focal disease, sub-diaphragmatic location, subcapsular location, proximity to high-risk areas (adjacent to large vessels or extrahepatic organs). Demographic and clinicopathologic variables were collected, including patient age, ethnicity, gender, body mass index [19], comorbidities, and past surgical history. Preoperative liver function and AFP levels were also recorded. The model for end-stage liver disease (MELD) score and Child–Pugh [20] class were used to categorize the severity of the liver disease. Operative characteristics reported included incision type, concomitant procedures, blood loss, and operative time. Inspection of the liver and abdomen for signs of cirrhosis or ascites was reported at the time of operation or by final pathology report. Tumor sizes, location by hepatic segment, number of lesions, and BCLC-stage group were also collected. All patients underwent a triphasic CT scan of the liver with intravenous contrast on post-operative day one to ensure complete ablation prior to discharge and every three months for the first year and six months thereafter to monitor for progression of disease, as previously described [7]. Outcome measures included readmissions, post-operative complications, progression-free survival (PFS), and overall survival (OS). Progression was defined as any documented evidence of active/growing local, regional, or distant disease diagnosed by triphasic CT scan or tissue biopsy (when available) after undergoing MWA. Recurrences were defined as ablation recurrence if ablation success was confirmed within 1 month or less of ablation and new disease was seen within 1 cm of the ablation defect during follow up [21]. Locoregional recurrence was defined as new disease found within the liver (ipsilateral and/or contra-lateral liver) during follow up. Distant recurrences were defined as new disease found beyond the primary tissue of origin (liver).

### 2.2. Operative Details and MWA Technique

All operations were performed under general anesthesia after obtaining informed consent. Open MWAs were conducted through midline or subcostal incisions for patients with extensive prior abdominal surgeries. For procedures performed laparoscopically, entry to the abdominal cavity was preferentially achieved by Hasson cut down and placement of a 12-mm trocar at the level of the umbilicus. One additional 5-mm trocar was placed to access the targeted tumor(s) and to ensure appropriate retraction of the liver off surrounding vital structures during ablation (i.e., stomach, colon, diaphragm). Intra-operative ultrasound was utilized in all cases to guide placement of biopsy and then MWA needle(s). A disposable 2-mm port was used as a sheath for the needle placement to assist in tumor targeting and to avoid track contamination. Hemostasis was meticulously ensured at the end of each ablation. The MWA ablations were performed with the Acculis system and then converted to the Solero system (Angiodynamics, Latham, NY, USA), both 2450-mHz ablation systems.

### 2.3. Statistical Analysis

All continuous variables were reported as median with interquartile range (IQR), while categorical variables were reported as “*n*” (%). Median PFS and OS were determined by Kaplan Meier survival analysis. Univariate statistical analysis was performed using the Cox proportional hazards model. The Cox proportional multivariate hazard model was used to determine independent prognosticators of PFS and OS of clinical covariates found significant at an alpha = 0.1 (*p* < 0.1) in univariate analyses. Statistical significance was defined as *p* < 0.05. Calculations were performed using IBM SPSS statistical software (IBM Corp. Released 2019. IBM SPSS Statistics for MacOS, Version 26.0. Armonk, NY, USA: IBM Corp).

## 3. Results

Out of 363 consecutive patients receiving surgical MWA, 184 patients (Table 1) from 2004 to 2023 were identified as having primary HCC on final pathology. These patients underwent 192 separate operative MWAs with a total of 215 tumors ablated. The median age of the cohort was 65.5 years (IQR, 33–86). Caucasian (81%) males (70%) represented the majority of the population. Underlying viral hepatitis (HBV or HCV) was present in 53% and the median Charlson-Comorbidity Index (CCI) was 7 (IQR, 3–9). Total abdominal hysterectomy (15%) and cholecystectomy (20%) were the most common prior abdominal surgeries. Trans-arterial embolization (i.e., TACE or TARE) before MWA was performed on 18%, and 15% had prior oncologic surgery (resection) for their HCC. The majority of patients had preserved liver function (median MELD score 17.4, IQR 10.7–40) without ascites (97%) and were CP-A (89%).

Laparoscopic MWA was performed in 88% of cases while 12% had open MWA. (Table 2). Two patients who received open MWA underwent concomitant procedures (one complex small bowel resection and one ventral hernia repair). Three laparoscopic patients also had biliary procedures at the time of MWA including cholecystectomy, cholangiogram, and lymphadenectomy. The most common location for MWA were tumors located in hepatic segment 8. The median size of the largest tumor was 3 cm (IQR, 1.0–6.8). Sixteen patients (9%) experienced post-operative complications. The median length of hospital stay was 1 day. One patient was readmitted within 30 days post MWA and one 90-day mortality was observed. Ablation recurrences occurred in 3 (1.6%) patients, local/hepatic recurrence occurred in 34%, at a median time of 19 months (9–18). Distant progression was noted in 8%.

The median follow up was 28.1 months (IQR, 6.4–170). Median PFS was 29.3 months (0.2–170) (Figure 1a) and OS was 44.2 months (2.8–170) (Figure 1b).

On univariate Cox proportional hazards analysis, male gender (HR 2.37, 95% CI 1.11–5.09, *p* = 0.03) and AFP ≥ 20 ng/mL were negative predictors of PFS (HR 3.73, 95% CI 1.71–8.13, *p* < 0.001), whereas increasing albumin was associated with prolonged PFS (HR 0.50, 95% CI 0.27–0.94, *p* = 0.04). Notably, the albumin to bilirubin (ALBI) ratio demonstrated a near-significant prediction of PFS with grade 1 scores protective for PFS (HR 0.13, 95% CI 0.15–1.08, *p* = 0.05) (Table 3).

Table 4 represents univariate analysis for factors predictive of OS. Patients who were CP-A experienced prolonged OS (HR 0.17, 95% CI 0.05–0.51, *p* = 0.002) compared to those in class B. Next, AFP ≥ 20 ng/mL (HR 2.79, 95% CI 1.27–6.13, *p* = 0.01), increased total bilirubin (HR 3.57, 95% CI 1.44–8.86, *p* = 0.006), and operative resection prior to MWA (HR 3.85, 95% CI 1.09–13.61, *p* = 0.04) were negatively associated with OS.

Table 5 lists the independent predictors of PFS and OS determined by multivariate Cox proportional hazards model. Again, increasing albumin had a protective effect for PFS (HR 0.45, 95% CI 0.21–0.94, *p* = 0.03), where a single-unit increase in preoperative albumin was associated with a 55% less chance of experiencing progression. Next, an increase in preoperative total bilirubin (HR 2.92, 95% CI 1.06–7.99, *p* = 0.04) and AFP values ≥ 20 ng/mL (HR 4.15, 95% CI 1.71–10.15, *p* = 0.002) demonstrate increased probability of experiencing progression. Furthermore, increasing preoperative total bilirubin and AFP ≥ 20 ng/mL were also independent predictors of OS (total bilirubin HR 3.62, 95% CI 1.21–10.81, *p* = 0.02 and AFP ≥ 20 ng/mL HR 2.87, 95% CI 1.12–7.35, *p* = 0.03).

## 4. Discussion

This report contributes long-term oncologic outcomes of primary HCC treated with operative MWA and highlights prognostic factors of PFS and OS when prior studies are scarce and have mainly focused on percutaneous approaches [22,23,24,25]. The most significant finding from this manuscript is that preoperative total bilirubin and AFP ≥ 20 ng/mL both independently predict PFS and OS after operative MWA. As more studies confirm operative MWA outcomes, these findings may be useful to properly risk-stratify and counsel patients with early to intermediate HCC unable to tolerate definitive resection or OLT.

Advancement in operative thermal ablation technology has provided an excellent treatment alternative for HCC patients with well-compensated cirrhosis and is safe and efficacious, offering low rates of morbidity and locoregional recurrence [14,15]. In agreement, this study found low rates of post-operative complications (9%), 90-day mortality (1.6%), and locoregional recurrence of 42% with a median follow up of 24.1 months (IQR, 6.4–170). While direct percutaneous ablations are effective and widely utilized, a laparoscopic surgical approach should be readily available as many HCC cases are not amenable to percutaneous ablation due to location near the liver dome, capsule, and surrounding structures [26,27,28,29,30]. Our institution preferentially performs minimally invasive MWA whenever feasible. Laparoscopic surgical MWA has the advantage of direct peritoneal viewing, enhanced staging via intraoperative ultrasound, mobilization of the liver, and hemostasis if required [14,31]. Despite this, advocacy for the laparoscopic approach to MWA has only recently occurred [30,32]. In our experience, the main issue remains the dual complexity of laparoscopic ablation approach and laparoscopic ultrasound skill that are required for successful ablation. To implement the benefits of laparoscopic ablation over the open approach, we developed and recommend a graded method for training in laparoscopic ablation and ultrasound skills (Figure 2).

Here we have observed several clinicopathologic characteristics to be significantly associated with survival after operative MWA that fit well within current literature. First, HCC is a diagnosis that is more prevalent in males, with a male to female ratio ranging from 2:1 to 4:1 [33]. Therefore, the finding of male gender to negatively affect PFS in this cohort is an expected outcome. This is also in agreement with other reported gender-based outcomes in HCC, as repeatedly, females tend to have better survival rates than males with HCC [34,35,36].

In a systematic review, Gupta et al. reported 26 out of 29 included studies found increased serum albumin to be associated with better survival in cancers of the gastrointestinal tract [37]. Albumin also was an independent predictor of PFS in this study with increasing values protective against progression. Similarly, in a study of percutaneous RFA and MWA in 137 HCC patients, serum albumin was also an independent predictor of OS [22]. Low serum albumin levels were also unfavorable toward RFS and OS in a propensity score matched analysis comparing RFA to MWA in tumors meeting Milan criteria [38]. These findings are of particular importance as serum albumin can be preoperatively optimized [39].

Tumor invasion of adjacent parenchyma and biliary structures may lead to hepatic congestion, liver dysfunction, and elevated total bilirubin [40]. Moreover, when considering that HCC often develops out of chronic inflammation and cirrhosis, rising total bilirubin may indicate worsening underlying disease. Our finding of elevated total bilirubin to negatively impact PFS and OS makes clinical and physiologic sense because total bilirubin is one component of CP and MELD scores and is, therefore, a surrogate of liver function. Despite this, to the authors’ knowledge, no other studies investigating operative MWA have reported elevated preoperative total serum bilirubin to negatively impact PFS and OS until now.

Recently, a novel albumin-bilirubin (ALBI) grade has emerged as a biomarker to assess outcome in HCC and has shown benefit when incorporated into traditional staging systems [41,42,43]. A study evaluating the ALBI grade found it to be an independent predictor of survival in a cohort of 567 advanced-stage HCC patients treated with sorafenib [44]. The ALBI has also predicted outcomes after percutaneous RFA and MWA. In another study assessing ALBI in early-stage HCC treated with RFA, ALBI was able to stratify outcomes within the same CP score [45]. One study of 442 patients who underwent resection or MWA for HCC reported that patients with higher ALBI scores (worse liver function) had a greater survival benefit following MWA compared to those who underwent resection [46]. Although the ALBI only reached a near-significant value in predicting PFS on univariate analysis within this cohort, the present study and those referenced above underline the importance of thorough objective estimates of hepatic reserve prior to MWA intervention for patients with HCC.

Historically, the CP classification (serum bilirubin, albumin, prothrombin time, severity of ascites, and encephalopathy) has been the most common assessment of hepatic function. Patients who met Child–Pugh A classification in this analysis had better OS than those in class B on univariate analysis. Similarly, Ryu et al. have also reported CP-A patients with up to seven criteria (including the sum of the largest tumor’s diameter in cm and the total number of tumors) to have better survival outcomes after operative MWA [17]. However, in another large single-center study evaluating operative MWA for HCC, Baker et al. did not observe significant differences in survival between these groups [15]. Although these findings remain relevant, it should be noted that the correlation of CP-A status with survival may simply reflect a concordant relationship with the lower grade of illness that CP-A patients have and not tumor biology.

The fetal-specific glycoprotein AFP, which is expressed by over 70% of HCC patients, is again a useful biomarker in the setting of operative MWA. AFP levels ≥ 20 ng/mL were also found to negatively predict PFS and OS in a multivariate analysis. Similarly, higher levels of AFP have correlated to survival outcomes in other studies including percutaneous MWA and operative MWA. Ma et al. reported AFP levels > 400 ng/mL to be unfavorable for PFS and OS after ultrasound-guided percutaneous MWA [23]. In another retrospective study, AFP levels < 20 ng/mL and those between 20–400 ng/mL were favorable for RFS and OS compared to those > 400 ng/mL [25]. Moreover, Cillo et al., in an analysis of 169 laparoscopic ablations of HCC, also found AFP > 400 ug/dL to be a significant predictor of survival along with age, presence of diabetes, and albumin ≤ 37 g/L. However, this study was only composed of 5% MWA cases [47]. Again, most recently, Ryu et al. found patients with AFP levels ≥ 100 ng/mL independently associated with worse OS [17]. These multiple findings of the tumor marker AFP correlating with PFS and OS suggest that the strongest predictor of these outcomes in a patient population undergoing local ablation will be tumor biology.

The current work should be carefully interpreted with respect to its limitations. As a single-institution study with limited sample size, the generalizability of these predictive PFS results should be considered in the context of findings from larger cohorts. Laparoscopic MWA was preferentially performed within this cohort and thus selection bias may be affecting these outcomes. Furthermore, the role of preoperative interventions (i.e., TARE or TACE) and their impact on change in preoperative AFP was not assessed and could have influenced these values. Despite these limitations, we have outlined several clinicopathologic characteristics predictive of important oncologic outcomes that fit well within current literature.

## 5. Conclusions

In summary, this single center experience confirms a preferentially laparoscopic MWA approach for HCC to be a viable option for treatment of early to intermediate stages of disease, offering low morbidity and mortality with good disease control. Surgeons should continue to adopt this technique in treating early- to intermediate-stage HCC patients who are not candidates for percutaneous approaches. Tumor biology remains crucial in determining prognoses for patients with HCC. Further, preoperative assessment of hepatic function including serum albumin and total bilirubin is essential prior to considering patients for operative MWA. Multi-institutional and prospective studies are needed to validate these outcomes in more robust patient populations.

## Figures and Tables

**Figure 1 cancers-15-01965-f001:**
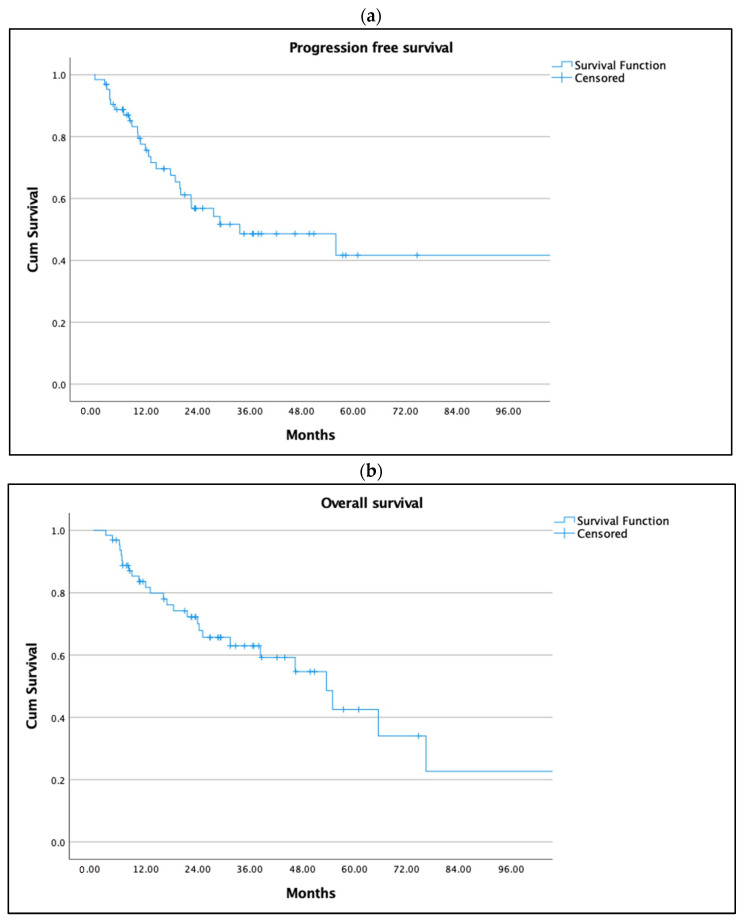
(**a**) Progression free survival and (**b**) overall survival of study cohort. All 184 patients were followed for PFS as presented (**a**) and OS as in (**b**).

**Figure 2 cancers-15-01965-f002:**
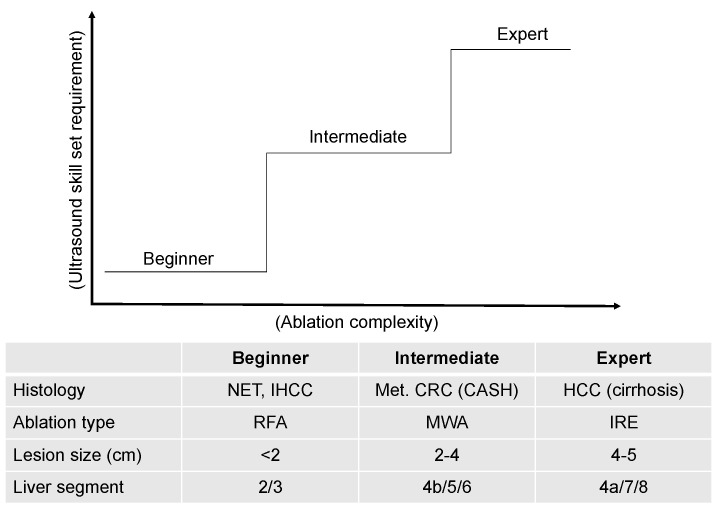
Graded method for educating laparoscopic ablation and ultrasound skill.

**Table 1 cancers-15-01965-t001:** Perioperative clinicopathologic details of the study cohort.

Characteristic	*n* = 184
Age, median (IQR)	65.5 (33–86)
Male gender, *n* (%)	45 (70)
BMI, median (IQR)	28.6 (17.9–42.6)
Ethnicity, *n* (%)	
Caucasian	149 (81)
African American	29 (16)
Other	6 (3)
Charlson-comorbidity index, median (IQR)	7 (3–9)
Cardiac	57 (31)
Pulmonary	35 (19)
Hepatic	129 (70)
Diabetes	66 (36)
Hepatitis	101 (55)
Alcohol use, *n* (%)	66 (36)
Tobacco use, *n* (%)	81 (44)
BCLC stage, *n* (%)	
0	15 (8)
A	147 (80)
B	22 (12)
Child–Pugh score, *n* (%)	
A	164 (89)
B	20 (11)
Prior surgery, *n* (%)	
Total abdominal hysterectomy	20 (16)
Cholecystectomy	35 (19)
Colectomy	15 (8)
Appendectomy	9 (5)
Orthopedic	35 (19)
None	29 (16)
Preoperative intervention, *n* (%)	
Prior surgery	9 (5)
Embolization	35 (19)
Ascites, *n* (%)	10 (6)
ALBI, *n* (%)	
Grade 1	82 (45)
Grade 2	82 (45)
Grade 3	5 (2)
Unknown	15 (8)
Laboratory Results	
Alpha fetal protein (AFP) ng/mL, median (IQR)	8.5 (0–79,560)
Platelet count, median (IQR)	141 (51–520)
Intervational Normalized Ration (INR), median (IQR)	1.1 (0.9–4.3)
Hemoglobin, median (IQR)	13.9 (9.5–18.5)
Albumin, median (IQR)	4 (2.1–5)
Total bilirubin, median (IQR)	0.8 (0.3–3.5)
Creatinine, median (IQR)	0.9 (0.5–1.7)
MELD Score, median (IQR)	17.4 (10.7–40)

IQR = inter-quartile range, BCLC = Barcelona clinic liver cancer, ALBI = albumin bilirubin grade, MELD = model for end stage liver disease.

**Table 2 cancers-15-01965-t002:** Operative and outcome details of the study cohort.

Characteristic	*n* = 184
Incision type, *n* (%)	
Laparoscopic	162 (88)
Open	22 (12)
Tumor number, median (IQR)	1 (1–10)
Size of largest tumor, cm, median (IQR)	3.0 (1.0–6.8)
Involved hepatic segments, *n* (%)	
I	0
II	11 (6)
III	11 (6)
IV	33 (18)
V	11 (6)
VI	40 (22)
VII	46 (25)
VIII	50 (27)
Concomitant procedures, *n* (%)	
Biliary (cholecystectomy, lymphadenectomy)	9 (5)
Small bowel resection	5 (2)
Ventral hernia	5 (2)
Estimated blood loss (mL), median (IQR)	50 (0–450)
Operative time (min.), median (IQR)	60 (35–170)
Length of stay (days), median (IQR)	1 (0–18)
Readmission <30 days, *n* (%)	1 (2)
Post-operative complications, *n* (%)	6 (3)
Local/Hepatic recurrence, *n* (%)	18 (34)
Progression free survival (months), median, (IQR)	29.3 months(0.2–170)
Overall survival (months), median, (IQR)	44.2 months (2.8–170).

**Table 3 cancers-15-01965-t003:** Prognostic factors of PFS univariate model.

Characteristic	Adjusted HR	95% CI	*p* Value
Age	1.01	0.97–1.06	0.59
BMI	0.99	0.92–1.07	0.77
Male gender	2.37	1.11–5.09	0.03
Ethnicity			
Other	REF		
African American	0.55	0.06–4.96	0.59
Caucasian	0.79	0.12–5.93	0.82
Charlson-comorbidity index CCI	1.25	0.95–1.64	0.12
Alcohol use	0.57	0.24–1.36	0.20
Tobacco use	0.84	0.38–1.84	0.66
BCLC stage			
0	REF		
A	1.92	0.27–13.70	0.51
B	2.11	0.49–9.10	0.32
Child–Pugh			
A	0.56	0.30–1.03	0.06
B	REF		
Hepatitis	0.63	0.29–1.34	0.24
Incision			
Laparoscopic	0.64	0.25–1.60	0.34
Open	REF		
Preoperative intervention			
None	REF		
Prior surgery	3.12	0.91–10.70	0.07
Embolization	0.84	0.31–2.26	0.73
Laboratory Results			
AFP (≥20 ng/mL)	4.16	1.71–10.15	0.002
Platelet count	1.00	0.99–1.00	0.44
INR	0.70	0.19–2.49	0.58
Hemoglobin	0.92	0.75–1.13	0.41
Albumin	0.50	0.27–0.94	0.03
Total bilirubin	2.24	0.86–5.86	0.10
ALBI			
Grade 1	0.13	0.15–1.08	0.05
Grade 2	0.36	0.05–2.85	0.34
Grade 3	REF		
EBL	1.00	0.99–1.00	0.58
Number of lesions	1.19	0.88–1.60	0.25
Tumor size	1.07	0.81–1.41	0.67
Lesion Type Invader Pusher Hanger	4.3 2.1 0.85	1.86–7.54 0.94–7.44 0.78–1.44	0.001 0.09 0.34
LOS	0.98	0.87–1.12	0.81
Complications	1.07	0.32–3.59	0.91

BMI = body mass index, IQR = inter-quartile range, BCLC = Barcelona clinic liver cancer, ALBI = albumin bilirubin grade, MELD = model for end stage liver disease, LOS = length of stay, EBL = estimated blood loss.

**Table 4 cancers-15-01965-t004:** Prognostic factors of OS univariate model.

Characteristic	Adjusted HR	95% CI	*p* Value
Age	1.01	0.97–1.06	0.55
BMI	0.92	0.84–1.00	0.06
Male gender	1.54	0.70–3.40	0.29
Ethnicity			
Other	REF		
African American	0.42	0.04–4.13	0.46
Caucasian	0.59	0.08–4.47	0.61
CCI	1.21	0.91–1.60	0.19
Alcohol use	1.11	0.48–2.61	0.80
Tobacco use	1.10	0.48–2.50	0.83
BCLC stage			
0	REF		
A	1.38	0.43–4.36	0.59
B	1.83	0.77–4.31	0.17
Child–Pugh			
A	0.17	0.05–0.51	0.002
B	REF		
Hepatitis	0.58	0.26–1.30	0.18
Incision			
Laparoscopic	0.61	0.20–1.87	0.40
Open	REF		
Preoperative intervention			
None	REF		
Prior surgery	3.85	1.09–13.61	0.04
Embolization	1.11	0.43–3.05	0.84
Laboratory Results			
AFP (≥20 ng/mL)	2.87	1.12–7.35	0.01
Platelet count	0.99	0.99–1.00	0.29
INR	0.93	0.33–2.63	0.89
Hemoglobin	0.94	0.75–1.18	0.60
Albumin	0.61	0.31–1.18	0.14
Total bilirubin	3.57	1.44–8.86	0.006
EBL	0.99	0.98–1.00	0.12
Number of lesions	1.09	0.80–1.48	0.58
Tumor size	0.99	0.75–1.30	0.93
Tumor Type Invader Pusher Hanger	4.23 0.98 0.97	1.66–8.66 0.94–1.4 0.95–1.27	0.01 0.31 0.44
LOS	0.93	0.81–1.07	0.35
Complications	0.50	0.12–2.18	0.36

BMI = body mass index, IQR = inter-quartile range, BCLC = Barcelona clinic liver cancer, ALBI = albumin bilirubin grade, MELD = model for end stage liver disease, LOS = length of stay, EBL = estimated blood loss.

**Table 5 cancers-15-01965-t005:** Independent predictors of PFS and OS Cox multivariate model.

Characteristic	Adjusted HR	95% CI	*p* Value
Progression free survival			
Gender	1.22	0.51–2.91	0.65
Child–Pugh	0.45	0.10–1.98	0.29
Preoperative intervention	0.61	0.31–1.20	0.15
Albumin	0.45	0.21–0.94	0.03
Total bilirubin	2.92	1.06–7.99	0.03
AFP (≥20 ng/mL)	4.16	1.70–10.15	0.002
Lesions Type (Invader)	3.16	1.91–9.15	0.002
Overall survival			
BMI	0.89	0.80–1.01	0.08
Child–Pugh	0.23	0.05–1.03	0.05
Preoperative intervention	0.78	0.36–1.67	0.52
Total bilirubin	3.62	1.21–10.81	0.02
AFP (≥20 ng/mL)	2.87	1.12–7.35	0.03
Lesion Type (Invader)	3.32	1.21–9.81	0.02

BMI = body mass index.

## Data Availability

The data presented in this study are available in this article on request to the corresponding author.

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
