# Peer review of "Patient Selection and Outcomes of Laparoscopic Microwave Ablation of Hepatocellular Carcinoma"

_cancers, 2023, doi:10.3390/cancers15071965_

Round 1

Reviewer 1 Report

It is suggested to draw some line diagrams to more intuitively show the relationship between the influencing factors and PFS and OS.

Author Response

Thank you for the review.  Please advise, I do not understand what you mean by "line diagrams"?  The Table 5 outlines all of the factors significant for PFS and OS.  I do not see where I can draw lines on a table?

Reviewer 2 Report

Laparoscopic microwave ablation (MWA) of hepatocellular carcinoma is un- derutilized and predictors of survival in this setting are not well characterized. Methods: The prognostic value of clinicopathologic variables was evaluated on progression-free survival(PFS) and overall survival(OS) by univariate and multivariate analyses. Results: 184 patients with HCC(me- dian age 66, [3386], 70% male) underwent laparoscopic MWA(N=162, 88% laparoscopic) compared to 12% undergoing open MWA(N=22). Median PFS was 29.3 months(0.2170) and OS was 44.2 months (2.8170). Ablation success was confirmed in 100% of patients. Ablation recurrence occurred
in 3%(6/184), local/hepatic recurrence occurred in 34%%, at a median time of 19 months (918). Distant progression was noted in 8%. Median follow up was 34.1 months (6.4 170). Procedure-related complications were recorded in 6 (9%) patients with one 90-day mortality. Further, >1 lesion, AFP levels 80 ng/mL, and a “invader” on pre-operative radiology were associated with increased risk of progression (>1 lesion HR 2.92, 95% CI 1.06 7.99, p=0.04, AFP 80 ng/mL HR 4.16, 95% CI 1.71 10.15, p=0.002, Invader HR 3.16, 95% CI 1.919.15, p=0.002 ) and mortality ( >1 lesion HR 3.62, 95% CI 1.2110.81, p=0.02], AFP 80 ng/mL HR 2.87, 95% CI 1.127.35, p=0.03, Invader HR 3.32, 95% CI 1.219.81, p=0.02). Conclusions: Preoperative lesion number, AFP 80 ng/mL, and a aggressive imaging characteristic(Invader) independently predict PFS and OS following laparoscopic operative MWA.

Figure 1. (a) Progression free survival and (b) overall survival of study cohort. Please, describe

The statistical analysis and characterization analysis is low

Author Response

Thank you for the review.  I am no sure what more needs to be described in Figure 1, but have added more text.  Can you please describe what you mean by "low".  Do you need more information about the statistical program?  Do Table 3 and 4 not demonstrate all the variables that we review??  Glad to update, but do not know where?

Reviewer 3 Report

You need to check again the conclusions in the abstract.

the AFP cut off levels are different between the abstract and the text .

also the number of the lesions and being invaders, ate not discussed at all in the manuscript?

Author Response

Thank you, these changes have been made and submitted with review

Reviewer 4 Report

Review Report

·     In this paper, the authors aimed to characterize predictors of survival after MWA of HCC.

·     Based on a prospective analysis of 184 patients, the authors concluded that preoperative lesion number, AFP ≥80 ng/mL, and an aggressive imaging characteristic (Invader) independently predict PFS and OS following laparoscopic operative MWA.

·     The paper is interesting. The authors have clearly worked hard to detail their study, but I have some comments:

POINTS OF STRENGTH

1.     Interesting topic.

2.     Prospective analysis

SPECIFIC COMMENTS

1.     The title should be corrected to “Patient Selection and Outcomes of Laparoscopic Microwave Ablation of Hepatocellular Carcinoma”

2.     The type of the study should be mentioned in the abstract and the main text.

3.     The aim of the study should be mentioned clearly in the abstract.

4.     Abstract: “aggressive imaging characteristic (Invader)…..What is meant by (Invader).

5.     What was the power of sample size calculation?

6.     Please clarify the inclusion and exclusion criteria of the study.

7.     Flow chart of the study and the number of excluded cases are required.

8.     What was the number of radiologists who reviewed the CT, and what was their experience?

9.     What was the number of interventional radiologists, and what was their experience?

10.  Footnotes below the tables are required.

Author Response

Thank you for your very helpful review:

  1. The title should be corrected to “Patient Selection and Outcomes of Laparoscopic Microwave Ablation of Hepatocellular Carcinoma”  *** Thank you, this was changed. ***
  2. The type of the study should be mentioned in the abstract and the main text. *** Added in both places ***
  3. The aim of the study should be mentioned clearly in the abstract.  *** Added***
  4. Abstract: “aggressive imaging characteristic (Invader)…..What is meant by (Invader).  *** we do not have enough characters to define this in the abstract, but provide extensive description in the methods. ***
  5. What was the power of sample size calculation?  *** This is a single arm study and was not compared to any other technique, so no this was not performed. ***
  6. Please clarify the inclusion and exclusion criteria of the study.  *** Thank you, added ***
  7. Flow chart of the study and the number of excluded cases are required.
  8. What was the number of radiologists who reviewed the CT, and what was their experience?  *** 4 body radiologist with extensive experience. ***
  9. What was the number of interventional radiologists, and what was their experience?  *** 4 IR with over 8 years of experience minimum. ***
  10. Footnotes below the tables are required. *** Added, thank you***

Round 2

Reviewer 4 Report

The authors performed a good job and responded to most of the reviewer's comments and improved the manuscript